# Optimal Shortcuts to Adiabatic Control by Lagrange Mechanics

**DOI:** 10.3390/e25050719

**Published:** 2023-04-26

**Authors:** Lanlan Ma, Qian Kong

**Affiliations:** International Center of Quantum Artificial Intelligence for Science and Technology (QuArtist), Department of Physics, Shanghai University, Shanghai 200444, China

**Keywords:** classical control, shortcuts to adiabaticity, inverse engineering, optimal control theory

## Abstract

We combined an inverse engineering technique based on Lagrange mechanics and optimal control theory to design an optimal trajectory that can transport a cartpole in a fast and stable way. For classical control, we used the relative displacement between the ball and the trolley as the controller to study the anharmonic effect of the cartpole. Under this constraint, we used the time minimization principle in optimal control theory to find the optimal trajectory, and the solution of time minimization is the bang-bang form, which ensures that the pendulum is in a vertical upward position at the initial and the final moments and oscillates in a small angle range.

## 1. Introduction

In the past few decades, with the development of experimental technology, the unstable control of nonlinear unstable systems [1,2,3,4,5] by a moving controller has attracted more and more attention. As far as we know, a sufficiently slow adiabatic process is a very simple way to achieve the ideal transfer, i.e., without excitations or losses [6,7,8,9]. However, a sufficiently slow process is problematic, firstly a sufficiently slow process, is impractically long and, secondly, over a long period of time, the ideal dynamics may be subject to the accumulation of random and/or uncontrollable perturbations that can undermine the expected results. In some experiments and theories, it is necessary to speed up this transport process but also to ensure that there is no excitation or loss at the initial and final moments [10,11,12,13,14,15,16,17]. To solve these problems, “Shortcuts To Adiabaticity” (STA) technology was proposed decades ago [18]. It achieves the same results as the adiabatic process in a short time [19]. At present, STA technology has been successfully applied to atoms, molecules, optical physics, solid state physics, chemistry, engineering, quantum systems, and classical systems [12,20,21,22,23,24,25]. With the development of STA technology, the STA technique includes many different schemes, such as the quantum invariant-based inverse control method [26,27,28,29], the reverse thermal transfer compensation method [30,31,32], the no-jump quantum drive method [33,34,35], etc. These methods have been validated in a number of experiments.

Cartpole is a typical multivariate, higher-order, nonlinear, strongly coupled, naturally unstable system. The stable control of the cartpole system is a typical problem in control theory. Here, we used the inverse engineering method, based on Lagrange mechanics, for a fast but stable control of the cartpole. By fixing the boundary conditions at the initial and final times, we aimed to find the trajectories that meet the boundary conditions [19,36] but that also achieves transition-free evolution ideally in a shorter time. Later, this approach can be complemented by optimal control theory [37,38], where we used the time minimization criterion in the Pontryagin maximization principle. Pontryagin’s maximum (or minimum) principle [39] is used in optimal control theory to find the best possible control for taking a dynamical system from one state to another, especially in the presence of constraints for the state or input controls. It was formulated in 1956 by the Russian mathematician Lev Semenovich Pontryagin and his students. The main thing is to find a controller so that the control Hamiltonian of the system reaches the maximum (or minimum) and then determines the optimal trajectory of the system. In this article, we used the relative displacement of the ball and the trolley horizontally as a controller, and then used the principle to find the optimal trajectory.

## 2. Physical Model

Here, we studied an ideal straight type cartpole system model; its schematic diagram and related parameters are shown in Figure 1. The trolley moves from $x_{0}=0$ and $x_{0}=d$, swinging the pole as far as possible in a vertically upward position at the initial and final moments. Never a perfect harmonic in actual transmission, similar to the transport potential of approximate optical tweezers, the quartic anharmonic plays a very important role, as shown in Figure 1b. We aimed to combine the inverse engineering method [26,27,28,29] with the optimal control theory in the Pontryagin maximization principle [37,38]. By setting boundary conditions to suppress the final excitation, we used the relative distance between the ball and the trolley as a controller throughout the process, to achieve the time minimization.

The physical model and related parameters are shown in Figure 1, where *M* indicates the mass of the trolley, *l* is the length of the non-elongated rod, *m* is the mass of the ball, θ is the angle between the pole and the vertical direction, $x_{0}$ represents the position of the trolley over time, and *x* represents the position of the ball. We assumed that the model is an ideal model by setting the following conditions: (i) the quality and friction of the rod can be ignored; (ii) point masses; (iii) constant rod length *l*; (iv) the trolley position is treated as a control parameter rather than a dynamical variable.

## 3. Hamiltonian of the System and Inverse Engineering

The solution of the cartpole system can be solved by Lagrange mechanics. According to the parameters provided in Figure 1, the Cartesian coordinate of mass in a rest frame are given by:(1)$$\begin{array}{c} x=x_{0}-l\sin\theta ,\\ y=l\cos\theta , \end{array}$$
where $x_{0}$ represents the position of the trolley. The kinetic energy and potential energy of the system can be expressed as follows:(2)$$\begin{array}{c} T=\frac{1}{2}M{\dot{x}}_{0}^{2}+\frac{1}{2}m\left({\dot{x}}^{2}+{\dot{y}}^{2}\right),\\ V=mgy, \end{array}$$
where the dots represent time derivatives. It can be obtained by substituting Equation (Equation 1) into Equation (Equation 2),
(3)$$\begin{array}{c} T=\frac{1}{2}\left(M+m\right){\dot{x}}_{0}^{2}+\frac{1}{2}ml^{2}{\dot{\theta }}^{2}-m{\dot{x}}_{0}l\dot{\theta }\cos\theta ,\\ V=mgl\cos\theta . \end{array}$$
Thus, the Lagrangian L=T−V of the system can be written as:(4)$$L=\frac{1}{2}\left(M+m\right){\dot{x}}_{0}^{2}+\frac{1}{2}ml^{2}{\dot{\theta }}^{2}-m{\dot{x}}_{0}l\dot{\theta }\cos\theta -mgl\cos\theta .$$
Based on the Euler–Lagrange equation $\frac{d}{dt}\left(\frac{\partial L}{\partial \dot{\theta }}\right)-\frac{\partial L}{\partial \theta }=0$, we can obtain the corresponding dynamics equation as follows:(5)$$\ddot{\theta }=\frac{g}{l}\sin\theta +\frac{{\ddot{x}}_{0}}{l}\cos\theta .$$
In order to reverse engineer the trajectory of the trolley, we need to calculate the Hamiltonian of the system, according to $p_{\theta }=\frac{\partial L}{\partial \dot{\theta }}$, where $p_{\theta }$ is the momentum with θ as the generalized coordinate. So, we need the conjugate momentum of θ,
(6)$$p_{\theta }=ml^{2}{\dot{\theta }}^{}-m{\dot{x}}_{0}l\cos\theta .$$
Thus, the Hamiltonian $H_{\theta }=\dot{\theta }p_{\theta }-L$ can be obtained from the Lagrangian: (7)$$\begin{array}{c} H_{\theta }=\frac{1}{2}ml^{2}{\dot{\theta }}^{2}-\frac{1}{2}\left(M+m\right){\dot{x}}_{0}^{2}+mgl\cos\theta . \end{array}$$
Using the canonical transformation approach, change the system from θ to *q* as the coordinate, and the Hamiltonian is rewritten as (please refer to Appendix A for details):(8)$$H=\frac{p^{2}}{2m}+mgl-\frac{1}{2}m\omega^{2}q^{2}-\frac{1}{8l^{2}}m\omega^{2}q^{4}-\frac{1}{2}\left(M+m\right){\dot{x}}_{0}^{2}-m{\ddot{x}}_{0}q,$$
where $\omega =\frac{g}{l}$ is the natural frequency of the pendulum, *p* represents the momentum with *q* as the generalized coordinate, $q=x_{0}-x$. According to the canonical transformation [40], we can obtain: (9)$$\begin{array}{c} \dot{q}=\frac{p}{m},\\ \dot{p}=m\omega^{2}q+\frac{m\omega^{2}}{2l^{2}}q^{3}+m{\ddot{x}}_{0}, \end{array}$$
from which the dynamic equation of the system becomes
(10)$$\ddot{q}-\omega^{2}q-\frac{\omega^{2}}{2l^{2}}q^{3}={\ddot{x}}_{0}.$$
This can be written as:(11)$$\ddot{x}+\omega^{2}\left(x_{0}-x\right)+\frac{\omega^{2}}{2l^{2}}{\left(x_{0}-x\right)}^{3}=0.$$
To fix the distance, we assumed that the trolley is transported from $x_{0}\left(0\right)=0$ to $x_{0}\left(t_{f}\right)=d$. We can set suitable boundary conditions for x(t) according to our expectations, and based on these boundary conditions the trolley trajectory $x_{0}\left(t\right)$ can be inverse engineered [26,27,28,29] to see how the position of the mass changes over time. We imposed the conditions as follows:(12)$$\begin{array}{c} x_{}\left(0\right)=0\begin{array}{cc} , & \end{array}x_{}\left(t_{f}\right)=d,\\ {\dot{x}}_{}\left(0\right)=0\begin{array}{cc} , & \end{array}{\dot{x}}_{}\left(t_{f}\right)=0,\\ {\ddot{x}}_{}\left(0\right)=0\begin{array}{cc} , & \end{array}{\ddot{x}}_{}\left(t_{f}\right)=0. \end{array}$$
In general, there are different options from which to choose the trajectories fulfilling the boundary conditions. For instance, one can choose the polynomial ansatz to interpolate the function. In the following section, we would like to complement it with time-optimal control for this problem.

## 4. Optimal Control for Time Minimization

In this section, we try to find the time minimization by using the Pontryagin maximum principle [37,38]. To be consistent, we set new notations for the Equation (Equation 11) of motion of the system:(13)$$y_{1}=x\begin{array}{cc} , & \end{array}y_{2}={\dot{y}}_{1}=\dot{x}\begin{array}{cc} , & \end{array}u\left(t\right)=x_{0}-x,$$
such that (where u(t) is the controller, representing the relative displacement of the ball to the trolley in the horizontal direction; *x* is the dynamic variable, representing the position of the ball)
(14)$$y_{2}={\dot{y}}_{1},$$
(15)$${\dot{y}}_{2}=-\omega^{2}u-\frac{\omega^{2}}{2l^{2}}u^{3}.$$

For this system, the boundary conditions for $y_{1}$, $y_{2}$ and u(t) can be given by Equations (Equation 11) and (Equation 12), and u(t) is required to satisfy u(t)=0 at t≤0 and $t\geq t_{f}$. From the optimal control theory, one can find $\left|u(t)\right|\leq \delta$ for a fixed bound δ and minimize the cost function *J*. The Pontryagin maximum principle finally provides the necessary conditions for optimal control [37,38].

For the relative displacement between the ball and the trolley, we utilized optimal control theory to find the time-optimal trajectory [38], so we defined the cost function as: $J=\int_{0}^{t_{f}}dt=t_{f}$. In this manner, we defined the control Hamiltonian as:(16)$$H_{c}\left(p\left(t\right),y\left(t\right),u\right)=p_{0}g\left(y\left(t\right),u\right)+p^{T}\cdot f\left(y\left(t\right),u\right).$$
For almost all $0\leq t\leq t_{f}$, the function $H_{c}\left(p\left(t\right),y\left(t\right),u\right)$ attains its maximum at $u=u^{*}$ and $H_{c}\left(p\left(t\right),y\left(t\right),u^{*}\right)=c$, where *c* is constant, and $\dot{y}=f\left(y\left(t\right),u\right)$. According to the cost function, the control Hamiltonian can be written as:(17)$$\begin{array}{c} H_{c}=p_{0}+p_{1}y_{2}+p_{2}\left[-\omega^{2}u-\frac{\omega^{2}}{2l^{2}}u^{3}\right], \end{array}$$
from which the canonical equations are
(18)$${\dot{p}}_{i}=-\frac{\partial H_{c}}{\partial y_{i}},$$
(19)$${\dot{y}}_{i}=\frac{\partial H_{c}}{\partial p_{i}}.$$
Accordingly, we can obtain the following costate equations by way of Equation (Equation 18):(20)$$\begin{array}{c} {\dot{p}}_{1}=0,\\ {\dot{p}}_{2}=-p_{1}. \end{array}$$
We can easily get $p_{1}=c_{1}$ and $p_{2}=-c_{1}t+c_{2}$, where $c_{1}$, $c_{2}$ are constant. In order to maximize the control of the Hamiltonian, we can obtain the optimal u(t) as a form of bang-bang control,
(21)$$u\left(t\right)=\left\{\begin{array}{cc} \begin{array}{c} 0,\\ -\delta ,\\ \delta ,\\ 0, \end{array} & \begin{array}{c} t\leq 0\\ 0<t<t_{1}\\ t_{1}<t<t_{f}\\ t\geq t_{f} \end{array} \end{array}\right.,$$
as shown in Figure 2, with only one intermediate switching time at $t_{1}$. Since the controller u(t) is in the form of a switch, the trajectory of the trolley $x_{0}\left(t\right)$ changes abruptly at $t=0^{+}$. By substituting u(t) into Equation (Equation 11) and combining the boundary conditions in Equation (Equation 12), we eventually obtain the trajectory of x(t),
(22)$$x\left(t\right)=\left\{\begin{array}{cc} \begin{array}{c} 0,\\ \frac{1}{2}\omega^{2}\delta t^{2}+\frac{1}{4l^{2}}\omega^{2}\delta^{3}t^{2},\\ d-\frac{1}{2}\omega^{2}\delta {\left(t-t_{f}\right)}^{2}-\frac{1}{4l^{2}}\omega^{2}\delta^{3}{\left(t-t_{f}\right)}^{2},\\ d, \end{array} & \begin{array}{c} t\leq 0\\ 0<t<t_{1}\\ t_{1}<t<t_{f}\\ t\geq t_{f} \end{array} \end{array}\right.$$
According to $u\left(t\right)=x_{0}-x$, the trajectory of the trolley $x_{0}\left(t\right)$ can be represented as:(23)$$x_{0}\left(t\right)=\left\{\begin{array}{cc} \begin{array}{c} 0,\\ \frac{1}{2}\omega^{2}\delta t^{2}+\frac{1}{4l^{2}}\omega^{2}\delta^{3}t^{2}-\delta ,\\ d-\frac{1}{2}\omega^{2}\delta {\left(t-t_{f}\right)}^{2}-\frac{1}{4l^{2}}\omega^{2}\delta^{3}{\left(t-t_{f}\right)}^{2}+\delta ,\\ d, \end{array} & \begin{array}{c} t\leq 0\\ 0<t<t_{1}\\ t_{1}<t<t_{f}\\ t\geq t_{f} \end{array} \end{array}\right.$$
Solving the system of Equations (Equation 14) and (Equation 15), we find the switching time $t_{1}$ and final time $t_{f}$ as follows:(24)$$t_{1}=\frac{t_{f}}{2},$$
(25)$$t_{f}=\frac{2l}{\omega }\sqrt{\frac{2d}{2l^{2}\delta +\delta^{3}}}.$$

Again, the trajectory of the trolley and the trajectory of the swing ball are shown in Figure 3. Through the above control, the ball can be oscillated in a small angle range.

After finding the optimal trajectory, we brought it into the exact Equation (Equation 5) and approximate dynamic equations. Clearly, short times imply larger transient angles, as shown in Figure 4. In addition, for the long time, we can see that there is basically no big difference in the change of angle over time, but the difference is noticeable only for the smaller time, $t_{f}=3\;s$. The main reason is that if we ignore the higher-order terms, the approximation deviates from the reality. At the same time, because the δ is the relative displacement between the ball and the trolley, the maximum value of which is the length of the pendulum, when the δ has a fixed value, the $t_{f}$ will have a lower bound. In other words, when δ reaches a certain value, the transmission time of the system is relatively small, the speed is relatively large, and the system may break down so that the control is not very good. According to the analysis of physical meaning, when δ=l, in this case, the pendulum is in a “lying flat” state, that is, in horizontal direction, the system basically breaks down completely. In short, we can determine whether the system breaks down based on the change in relative displacement with δ.

## 5. Conclusions

In conclusion, we applied reverse engineering based on Lagrange mechanics and time minimization in the Pontriagin maximization principle for the fast but stable control of nonlinear unstable systems, e.g., the cartpole. In such control, we took an anharmonic effect and analyzed the stability of control. By fixing the relative displacement between the position of the ball and the trolley as constraints, we applied optimal control theory to design a time-optimal trajectory, as compared to the case of harmonic approximation. In the future, we will work out the classical control of the inverted double pendulum, and compare it with the control of the cartpole. Of course, the machine learning, fed by the results of STA, can be an interesting area for exploration as well.

## Figures and Tables

**Figure 1 entropy-25-00719-f001:**
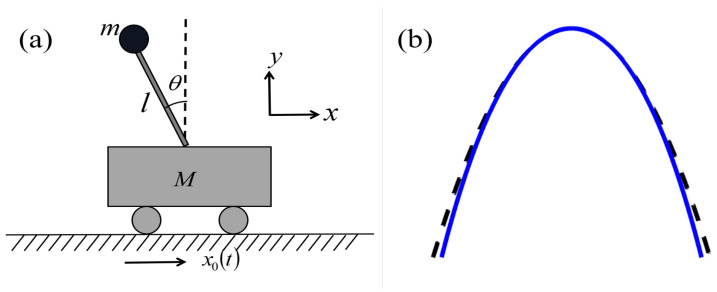
(**a**) Physical model of the cartpole and related parameters, *M* indicates the mass of the trolley, *l* is the length of the non-elongated rod, *m* is the mass of the ball, θ is the angle between the pole and the vertical direction. (**b**) Quartic potential, i.e., anharmonic potential (blue solid line) compared with quadratic potential, i.e., harmonic potential (black dash line).

**Figure 2 entropy-25-00719-f002:**
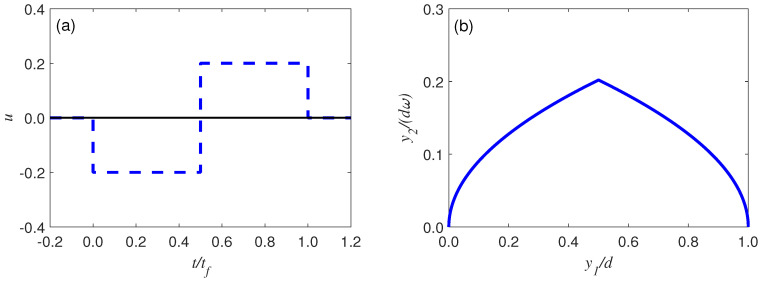
(**a**) The control function (dashed blue line) for the time–optimal problem, and (**b**) the corresponding trajectory (solid blue line) for $g=9.8\;ms^{-2}$, l=1m, d=5m, $\omega =\sqrt{\frac{g}{l}}$, δ=0.2m, $t_{f}=\frac{2l}{\omega }\sqrt{\frac{2d}{2l^{2}\delta +\delta^{3}}}$.

**Figure 3 entropy-25-00719-f003:**
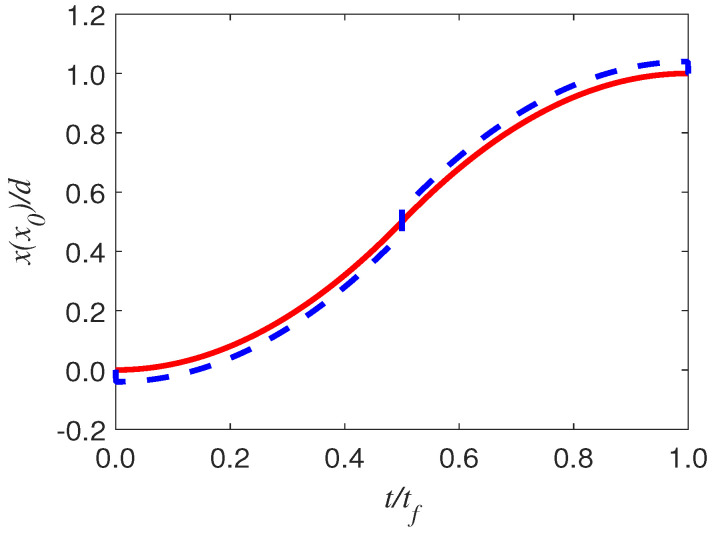
(Blue dash line) The trajectory of the trolley for time–optimal problem, and (red solid line) the corresponding trajectory of the mass for $g=9.8\;ms^{-2}$, l=1m, d=5m, $\omega =\sqrt{\frac{g}{l}}$, δ=0.2m, $t_{f}=\frac{2l}{\omega }\sqrt{\frac{2d}{2l^{2}\delta +\delta^{3}}}$.

**Figure 4 entropy-25-00719-f004:**
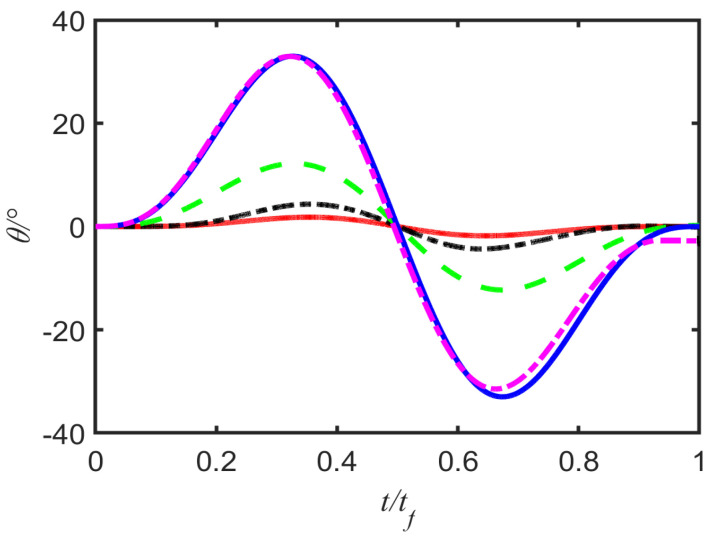
Graph of pendulum angle over time of this system for comparison of angle under exact dynamics with the angle in approximate cases; when $t_{f}=5\;s$ (dashed green line), $t_{f}=10\;s$ (dashed–dotted black line), $t_{f}=15\;s$ (solid red line), the exact and approximate curves essentially overlap perfectly. However, when $t_{f}=3\;s$, the exact (solid blue line) and approximate (dashed–dotted magenta line) curves are distinguishable. Other parameters: $g=9.8\;ms^{-2}$, l=1m, d=5m, $\omega =\sqrt{\frac{g}{l}}$, $t_{f}=\frac{2l}{\omega }\sqrt{\frac{2d}{2l^{2}\delta +\delta^{3}}}$.

## Data Availability

Not applicable.

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
