# Peer review of "Optimal Shortcuts to Adiabatic Control by Lagrange Mechanics"

_entropy, 2023, doi:10.3390/e25050719_

Round 1

Reviewer 1 Report

See attached file.

Author Response

We gratefully thank the editor and all reviewers for their time spend making their constructive remarks and useful suggestions.Please see the attachment for specific modifications.We sincerely hope that this revised manuscript has addressed all your comments and suggestions. We appreciated for reviewers’ warm work earnestly,and hope that the correction will meet with approval.Once again,thank you very much for your comments and suggestions.

Reviewer 2 Report

The present manuscript discusses the use of optimal control theory for designing optimal trajectories of a classical mechanical system (the transport of an inverted pendulum). The introduction provides sufficient background, the cited references are relevant and the research design is adequate.

However, I have some concerns about the work. Some are major concerns that the authors should address in order to reconsider my opinion on their work, while some other are minor comments and suggested changes.

See attached pdf with my main concerns and full report.

Author Response

(The authors gave the same response as above.)

Reviewer 3 Report

The authors studied a quite well-known model to optimize the trajectory and to transport the cartpole in a fast and stable way. It is more related to how to optimize an engineering problem, and little related to the concept of shortcuts to adiabaticity (STA). At least I did not see any strong motivations in the introduction. 

The proposed methods are quite common to be used for solving a closed quantum system, e.g. harmonic oscillator. As I known, this paper is similar to the harmonic oscillator with cubic anharmonicity (Eq. (9)). With the cubic anharmonicities, it already has been solved in the past. 

In summary, I think the current version of this paper is not suitable to be published in journal "Entropy". 

Author Response

(The authors gave the same response as above.)

Round 2

Reviewer 1 Report

Authors improved the quality of presentation, thus I assume this article can be published in its present form.

Author Response

It is a great honour to receive your approval of our work. Thank you for your hard work and good luck in your endeavours.

Author Response

We gratefully thank you for your time spend making their constructive remarks and useful suggestions.Please see the attachment for specific modifications.We sincerely hope that this revised manuscript has addressed all your comments and suggestions. We appreciated for reviewers’ warm work earnestly,and hope that the correction will meet with approval.Once again,thank you very much for your comments and suggestions.

Reviewer 3 Report

Thanks for authors' kind reply. The manuscript has been improved a lot except tons of typos in the reference section. I found some parts of the manuscript to be difficult to understand due to the language used, for example, the sentence "...initial and eventually moments" should change into "...initial and final moments" in Line 50.

It might be helpful to improve the written English to make the manuscript more accessible to readers.

Author Response

We gratefully thank you for your time spend making their constructive remarks and useful suggestions.Thanks to your valuable suggestions and comments, which improve our manuscript a lot. At the same time, we apologize for our writing quality, which did not meet his/her expectation, and we have had our manuscript touched up by professional.

We sincerely hope that this revised manuscript has addressed all your comments and suggestions. We appreciated for reviewers’ warm work earnestly,and hope that the correction will meet with approval.Once again,thank you very much for your comments and suggestions.

Round 3

Reviewer 2 Report

Please find attached my report.

Author Response

(The authors gave the same response as above.)
